# Enhancement of Photoluminescence Quantum Yield and Stability in CsPbBr_3_ Perovskite Quantum Dots by Trivalent Doping

**DOI:** 10.3390/nano10040710

**Published:** 2020-04-09

**Authors:** Sujeong Jung, Jae Ho Kim, Jin Woo Choi, Jae-Wook Kang, Sung-Ho Jin, Youngho Kang, Myungkwan Song

**Affiliations:** 1Surface Technology Division, Korea Institute of Materials Science (KIMS), 797 Changwondae-ro, Sungsan-Gu, Gyeongsangnam-do, Changwon 51508, Korea; sujeong2@kiost.ac.kr (S.J.); jho83@kims.re.kr (J.H.K.); jinwoo.choi@kims.re.kr (J.W.C.); 2Department of Flexible and Printable Electronics, Polymer Materials Fusion Research Center, Chonbuk National University, Jeonju 54896, Korea; jwkang@jbnu.ac.kr; 3Department of Chemistry Education Graduate Department of Chemical Materials Institute for Plastic Information and Energy Materials, Pusan National University, Busan 46241, Korea; 4Materials Data Center, Korea Institute of Materials Science (KIMS), 797 Changwondae-ro, Sungsan-Gu, Gyeongsangnam-do, Changwon 51508, Korea; 5Department of Materials Science and Engineering, Incheon National University, Incheon 22012, Korea

**Keywords:** CsPbBr_3_, perovskite, nanocrystals, trivalent doping, ligand-assisted reprecipitation

## Abstract

We determine the influence of substitutional defects on perovskite quantum dots through experimental and theoretical investigations. Substitutional defects were introduced by trivalent dopants (In, Sb, and Bi) in CsPbBr_3_ by ligand-assisted reprecipitation. We show that the photoluminescence (PL) emission peak shifts toward shorter wavelengths when doping concentrations are increased. Trivalent metal-doped CsPbBr_3_ enhanced the PL quantum yield (~10%) and air stability (over 10 days). Our findings provide new insights into the influence of substitutional defects on substituted CsPbBr_3_ that underpin their physical properties.

## 1. Introduction

Lead halide perovskites (LHPs) are emerging semiconductors with potential applications in energy and optoelectronic devices, such as displays [1], lasers [2,3,4], solar cells [5,6,7,8,9], and light-emitting diodes (LEDs) [10,11]. A low-cost and facile process can be used to fabricate LHP films for high-performance devices, which offers advantages over well-known semiconductors, e.g., Si and chalcogenide compounds. To date, organic–inorganic hybrid materials, such as CH_3_NH_3_PbX_3_ (MAPbX_3_, X = Cl, Br, and I), have received the most focus in the literature. However, devices using hybrid perovskites lack stability as the organic components are vulnerable to oxygen and moisture [12,13,14,15,16,17]. All-inorganic LHPs, wherein cesium replaces the molecular cation (CsPbX_3_), exhibit superior stability and similar optoelectronic properties compared to the hybrid LHPs, and thus, have been considered as alternatives [18,19,20].

Nanosized CsPbX_3_ quantum dots (QDs) are of particular interest as the active layer in visible-light emitting devices, owing to the band gaps (1.7 to 3.0 eV) that can be tuned by the particle size, composition [21], crystal dimension [22,23,24,25,26], and their emission spectra can cover a full range of visible light [27,28]. In addition, CsPbX_3_ QDs exhibit both high photoluminescence quantum yields (PLQYs)—up to 90%—and sharp emission linewidths of 12–42 nm, simultaneously. This is achieved without the formation of complex core-shell structures, which are required for chalcogenide-based QDs. Despite the potential of CsPbX_3_ QDs, there remain two challenges for practical utilization. First, PLQYs larger than 90% have been achieved for the red spectral ranges, but they decrease when the spectral range shifts to higher frequencies and is limited to less than 50% in the violet. Low PLQY for high-frequency visible light hampers the applicability of CsPbX_3_ QDs for various applications. Second, the PL brightness of the QDs diminishes when exposed to ambient air with complete degradation within a few days. This lack of stability must be overcome for commercial applications.

Impurity doping has been recognized as a feasible route to modulate the structural, optical, and electrical properties of LHP QDs. Yong and coworkers demonstrated that Ni^2+^ doping enhances the short-range lattice order of CsPbCl_3_ QDs, which removes defect states in the lattice, resulting in an increase in the PLQY of up to 96.5% for the violet emission [29]. Zou et al. reported that substituting Pb^2+^ with Mn^2+^ enhanced the thermal stability of CsPbBr_3_ QDs, enabling LED fabrication with a higher external quantum efficiency (EQE) under ambient conditions [30]. Doping of heterovalent Bi^3+^ into CsPbBr_3_ QDs was also investigated, which resulted in shifts of the highest occupied molecular orbital (HOMO) and lowest unoccupied molecular orbital (LUMO) levels of the QD, thereby promoting charge transfer with molecular acceptors at the interface [31].

However, few studies have reported simultaneous improvements in PLQY and long-term stability [32,33]. In this work, we show that doping with trivalent elements can concurrently enhance both properties of CsPbX_3_ QDs. Specifically, we investigate the LHP, CsPbBr_3_. Our results show that the PLQY of M^3+^-doped CsPbBr_3_ QDs (M = In, Sb, and Bi) initially increases with the doping concentration. It then reaches a maximum and drops with increasing doping content. Overall, Sb and In enhance the PLQY more than Bi, leading to ~90% of the maximum PLQY. Results from density functional theory (DFT) calculations provide an explanation on the observed PLQY enhancement due to dopant inclusion. Finally, we show that M^3+^-doped QDs maintains the high PL intensity over an extended period of time compared to undoped QDs.

## 2. Materials and Methods

### 2.1. Chemicals

All reagents were used without any purification: PbBr_2_ (lead dibromide, ≥98%), InBr_3_ (indium tribromide, 99%), oleic acid (technical grade, 90%), butylamine (99.5%), toluene (ACS (American Chemical Society) reagent, ≥99.5%), acetonitrile (anhydrous, 99.8%), and DMF (N,N-dimethylformamide, anhydrous, 99.8%) were purchased from Sigma-Aldrich (St. Louis, MO, USA). CsBr (cesium bromide, 99.9% metals basis), SbBr_3_ (antimony tribromide, 99%) and BiBr_3_ (bismuth tribromide, 99% metals basis) were purchased from Alfa Aesar (Haverhill, MA, USA).

### 2.2. Synthesis of Quantum Dots

Colloidal CsPbBr_3_ QDs were synthesized by ligand-assisted re-precipitation (LARP) method at room temperature, as illustrated Appendix A. To be specific, 0.4 mmol of CsBr and 0.4 mmol of PbBr_2_ were first dissolved in 10 mL N,N-dimethylformamide (DMF) or dimethyl sulfoxide (DMSO). For doping of the trivalent elements, 1, 3, 5, and 10 mol% of MBr_3_ (M = In, Sb and Bi) were added in this solution. Then 1 mL of oleic acid and 0.5 mL of butylamine were added to form CsPbBr_3_ precursor solution. For the formation of perovskite QDs, 150 μL of the precursor solution were dropped into 10 mL of toluene under vigorous stirring at room temperature. Afterward, solution was centrifuged at 8000 rpm for 3 min to enhance the uniformity of the quantum dots size. Finally, acetonitrile was added to the solution to remove precipitated particles and organic matters and further centrifuged at 9000 rpm for 5 min. To equalize environmental conditions, all experiments were conducted in the hood with both an average temperature of 20 °C and a humidity of 40%.

### 2.3. Measurement and Characterization

The optical absorption and PL spectra were measured for the CsPbBr_3_ QDs in solution. The absorption spectra were recorded using a ultraviolet–visible–near infrared (UV–Vis–NIR) spectrophotometer (Agilent technologies, Cary 5000, Santa Clara, CA, USA). The PL spectra were carried out using an absolute PLQY spectrometer (Hamamatsu photonics, Quantaurus-QY, Hamamatsu, Japan) under UV (400 nm) illumination with a 150 W xenon lamp. The time-resolved PL decay was recorded using Fluorescence lifetime analysis (Hamamatsu photonics, Quantaurus-tau, Hamamatsu, Japan) equipped with a 405 nm, 200 kHz pulse. The high-resolution transmission electron microscope (HRTEM) image of QDs were obtained using Hitachi, HF-3300. Samples for TEM were prepared by dropping of the CsPbBr_3_ QDs in toluene onto a carbon-coated 300 mesh copper grid with support films. The X-ray diffraction (XRD) measurements were carried out using Panalytical, X’pert-PRO (Almelo, Netherlands). For the XRD measurements, the solution of CsPbBr_3_ QDs were dropped into a cleaned glass, and then dried at room temperature for 30 min. The HOMO level were measured by UV photoelectron spectroscopy (UPS) using ULVAC-PHI. The x-ray photoelectron spectroscopy (XPS) measurements were conducted employing PHI 5000 Versa Probe II.

### 2.4. Calculation

Our defect analysis is based on DFT calculations using Vienna ab initio simulation package (VASP) with projector augmented waves (PAW) [34]. The cutoff energy of 300 eV was used for expanding the plane-wave basis and only Γ point was sampled for the Brillouin-zone (BZ) integration. For the exchange-correlation energy, we employed the PBE and HSE06 hybrid functional [35,36,37]. Throughout our calculations, the spin-orbit coupling (SOC) effects that are known to be crucial for correctly producing the electronic structure of CsPbBr_3_ due to the presence of Pb were included. The 50% mixing of the exact-exchange energy was used for the HSE06 hybrid functional calculations. The HSE06 + SOC calculation yields the band gap of 2.11 eV of the bulk orthorhombic phase of CsPbBr_3_, which is in good agreement with experiments [20].

The defect formation energy *E*^f^ is computed as follows:(1)$$E^{f}\left(D^{q}\right)=E_{\mathrm{tot}}\left(D^{q}\right)-E_{\mathrm{tot}}\left(\mathrm{clean}\right)-\underset{i}{\sum }n_{i}\mu_{i}+qE_{F}+\Delta_{q},$$
where *E*_tot_ (*D^q^*) and *E*_tot_ (clean) are the total energy of a supercell with a defect *D^q^* in charge state q and the perfect supercell, respectively. *n_i_* is the number of atoms with type *i* which is removed from (*n_i_* < 0) or added into (*n_i_* > 0) the supercell and *μ_i_* is their chemical potential. *μ_i_* reflects the growth conditions and should be limited to ensure the phase stability (see Appendix A). The Fermi level (*E*_F_) is the chemical potential of electrons, which is referenced to the valence band maximum. ∆*_q_* is a correction term arising from the finite supercell size, which is evaluated by the method of Freysoldt et al. [38].

## 3. Results and Discussion

Pure CsPbBr_3_ QDs (PQDs) have multiple structures that exist between a circular and cubic morphology, with an average particle size of 5.0 ± 1.0 nm (Figure 1a). The XRD patterns (Figure 1b) are consistent with results in the previous work. While the crystal structure cannot definitively be assigned [39], two distinct peaks are observed, which correlate well with the (110) at 15.26° and (220) at 30.75° crystallographic planes of the perovskite orthorhombic structure (ICDD 98-9-7851) and the (200) at 15.12° and (400) at 30.46° those of the cubic structure (ICDD 1-72-7930) [40]. The orthorhombic phase, containing a slight distortion of the PbBr_6_ octahedron, is known to be more stable than the cubic phase in bulk at room temperature. Therefore, mixed phased crystal was formed during the synthesis. According to Bragg’s law, *nλ = 2dsinθ*, the lattice distance in the (220) direction is 0.27 nm, which was further confirmed by high-resolution transmission electron microscopy (HR-TEM) (Figure 1a). A doping concentration of 3 mol% was used for In- and Sb-doped CsPbBr_3_ QDs (IPQDs and SPQDs) and 1 mol% for Bi-doped CsPbBr_3_ QDs (BPQDs). These doping concentrations yielded the highest PLQYs. However, doping with trivalent elements does not alter the QD shape or size (Figure 1a). Furthermore, no peak shifts in the XRD patterns were observed (Figure 1b), suggesting that dopant incorporation has no effect on the crystal structure. This finding is consistent with 2D in situ Grazing Incidence Wide Angle X-Ray Diffraction (GIWAXD) analysis (Pohang Accelerator Laboratory (PAL), Appendix A), showing no significant change in the crystal structure. However, changes in the intensity of the XRD peak between the PQD and IPQD, SPQD, and BPQD was observed. The intensity of the (00c) planes increased after doping, as it caused growth in the out-of-plane direction for the (00c) plane.

The XPS results in Appendix A illustrate that the PQD and doped QDs consist of Cs, Pb, Br, C, and O [41]. The XPS profile of IPQDs shows two additional peaks at 445.8 and 453.5 nm compared to that of PQDs (Appendix A), which are attributed to the In 3d_5/2_ and In 3d_3/2_ components, respectively. For SPQDs, there are two peaks with binding energies of 530.8 and 540.5 eV that are attributed to Sb 3d_5/2_ and Sb 3d_3/2_ (Appendix A), respectively. In the case of BPQDs, there are two peaks associated with Bi 4f_7/2_ and 4f_5/2_ core levels at 159.0 and 164.3 eV for BPQDs (Appendix A) [42]. These XPS results confirm that dopants are successfully incorporated into the QDs.

Upon comparing the optical properties of the PQDs and doped QDs, it is observed that absorption onset of PQD appears at ~513 nm, indicating a band gap of 2.42 eV (Figure 2a). PQDs display a sharp PL peak at 513 nm with a narrow full width at half maximum (FWHM) of 22 nm. Doping results in a small blue-shift in the absorption onset or PL peak position up to 0.06 eV (The absorption and PL spectra for IPQD, SPQD, and BPQD depending on the doping concentration are presented in Appendix A). The slight blue-shift in the absorption onset is attributed to an interaction between the conduction bands with impurity states of higher energy [31]. The PL peak positions of IPQD, SPQD, and BPQD are 509, 510, and 511 nm, respectively, confirming the blue-shift from PQD. Figure 2b shows a change in the PLQYs of CsPbBr_3_ QDs with respect to doping concentrations. The PLQY of PQD is 79% [31]. When In or Sb is introduced into QDs, the PLQY is found to initially increase with the doping concentration, exhibiting a maximum PLQY of 88.8% for IPQD and 91.2% for SPQD at 3 mol% doping (Table 1). The PLQY drops to ~82% with further doping to 10 mol%. Decrement on the perovskite crystallinity due to the excessive amount of dopants could be the origin of the PLQY drops in the high concentration of doping ratio. A small enhancement is observed in the PLQY of BPQD when doped at 1 mol%. However, Bi doping significantly deteriorated the PLQY to 57.2% at 10 mol%.

The doping-induced increase in PLQY is due to the enhanced confinement of photocarriers. Thin PbBr_x_ layers form at the PQD surface, resulting in a core–shell-like structure [27]. Since PbBr_x_ has a larger band gap than CsPbBr_3_, a quantum-well-like band alignment appears. Accordingly, the confinement of photocarriers within the PQDs becomes stronger, enhancing the incidence of radiative recombination. The DFT calculations show that MBr_3_ (M = In, Sb, and Bi) has a greater band gap than PbBr_2_ (Appendix A). The quantum-well effect is expected to be stronger when the dopants are incorporated at the QD surface considering the Pb-rich composition at the surface (Appendix A).

Dopants can also alter the relaxation dynamics of photocarriers, contributing to the change in the PLQY. Time-resolved PL measurements using time-correlated single photon counting spectroscopy were performed to determine the photophysical properties of QDs. The decay curves of the PL intensity for optimally doped QDs are plotted in Figure 3a (the evolution of the decay profile with respect to the doping density of Bi is present in Appendix A). The PL decay for IPQD and SPQD is slower than that of PQD. The photocarrier lifetime for IPQD and SPQD is 6.9 ns and 6.6 ns, respectively. These lifetimes are larger than that of PQD of 6.3 ns. This finding suggests that In and Sb suppress nonradiative recombination, yielding a longer lifetime, and subsequent PLQY enhancement. In contrast, the lifetime of photocarriers doped with 1 mol% Bi did not significantly change. The enhanced PLQY in doped QDs is retained for a longer period of time compared to PQDs, suggesting a better stability under ambient conditions (Figure 3b). The enhanced stability of the doped QDs may be associated with the suppression of ionic migration [43]. Incorporation of trivalent cations can reduce the concentration of native donors due to an increase in the Fermi level. Ionic diffusion is typically vacancy-mediated, and thus the diffusion process would effectively be suppressed.

DFT calculations were conducted to ascertain the impact of the dopants on PL relaxation. The HSE06+SOC method was used, which yields accurate band structures for LHPs. Explicit simulations of quantum dot structures of more than thousands of atoms are not feasible because of the extremely high computational cost. Thus, for simplicity, we used 2 × 2 × 2 supercells of the bulk orthorhombic phase (160 atoms) to simulate defects. Earlier work has shown that trivalent dopants, such as Bi can be a substitute for Pb in CsPbBr_3_. Therefore, we considered substitutional defects (MPb where M = In, Sb, and Bi), as depicted in Figure 4a. Detailed atomic and electronic structures of each dopant are provided in Appendix A, respectively. Point defects can act as a recombination center for charge carriers, deteriorating the PLQY of materials. This defect-induced recombination, i.e., Shockley–Read–Hall (SRH) recombination, typically occurs non-radiatively without the emission of a photon [44]. The rate of the SRH recombination depends upon the energetic position of the charge transition level, ε(*q*/*q*’), the of defects, which is given by:(2)$$\varepsilon \left(q/q^{\prime }\right)=\frac{E^{f}\left(D^{q^{\prime }},E_{F}=\mathrm{VBM}\right)-E^{f}\left(D^{q},E_{F}=\mathrm{VBM}\right)}{q-q^{\prime }}$$
where *E*^f^ (*D^q^*, *E*_F_ = valence band maximum (VBM)) denotes the defect formation energy of a defect in the charge state, *q*, when the Fermi level (*E*_F_) lies at the VBM. The charge transition level can be interpreted as a thermodynamic defect level (TDL) determined from the sum of the atomic relaxation energy and the electronic transition energy [38]. In accordance with the SRH model, the overall rate of SRH recombination increases as the TDL approaches the mid gap. The SRH recombination involves a capture process of one carrier (electron or hole), followed by recombination of the other carrier. When a TDL of a defect lies near the conduction or valence band edges, such a defect can quickly capture the corresponding carrier—e.g., electron for conduction band and hole for valence band—while the capture process of the other carrier occurs very slowly. Due to the TDL dependence of the overall SRH recombination rate, a defect with a TDL close to the mid-gap region causes a more significant loss of PLQY compared to others with a TDL around the band edges. Figure 4b displays the position of ε(1 + /0) for In_Pb_, Sb_Pb_, and Bi_Pb_. We note that ε(1 + /0) of Bi_Pb_ is closer to the mid gap (Appendix A) than that of In_Pb_ or Sb_Pb_, suggesting that Bi_Pb_ is a more effective SRH recombination center. These results support the significant loss of PLQY for the BPQD with high Bi doping densities. In_Pb_ and Sb_Pb_ can capture photocarriers non-radiatively, even if the effect per defect is expected to be weaker than Bi_Pb_. Therefore, significant doping of In and Sb is disadvantageous for the PLQY.

The longer photocarrier lifetime of the 3 mol% IPQDs or SPQDs is associated with the suppression of the formation of intrinsic recombination centers. Specifically, through examination of TDLs of native defects (Appendix A), we find that Pb_Br_ and Br_Pb_ can become critical recombination centers, as they develop defect levels close to the mid gap. Pb_Br_ is a donor whose formation energy increases with the Fermi level. UPS measurements show a slight increase in the Fermi level in doped QDs, particularly SPQD (Appendix A), because the dopants are single donors. However, the Fermi level, whose doping concentration is not significant, is compared with the doping level. The slight difference might be originated to the effective neutralization of the trivalent cations by halide anions in the perovskite structure. As a result, the concentration of Pb_Br_ is expected to be lower in the doped QDs compared to that in the PQD, decreasing the SRH recombination.

## 4. Conclusions

In conclusion, we discussed the influence of the dopants (In, Sb, and Bi) on the characteristics of CsPbBr_3_ QDs. The PLQY improved via incorporation of dopants at low concentration. Doping concentration of 3 mol% In, 3 mol% Sb, and 1 mol% Bi led to the highest efficiency. In addition, the CsPbBr_3_ perovskite QDs with optimal doping concentration showed longer lifetime and improved stability compared to PQDs. We confirmed that the incorporation of the trivalent cations can reduce the concentration of native donors due to an increase in Fermi level. Ionic diffusion is typically vacancy-mediated, thus the diffusion process is effectively suppressed.

## Figures and Tables

**Figure 1 nanomaterials-10-00710-f001:**
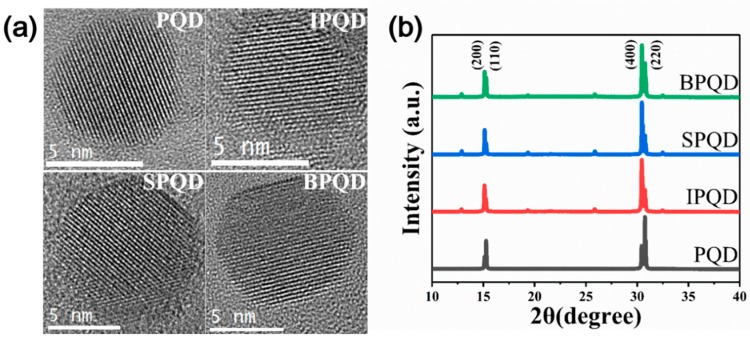
(**a**) High-resolution transmission electron microscopy (HR-TEM) images (**b**) X-ray diffraction (XRD) data of CsPbBr_3_ via ligand-assisted re-precipitation (LARP) method. Characterization of trivalent ion substituted CdPbBr_3_ perovskite quantum dots (QDs).

**Figure 2 nanomaterials-10-00710-f002:**
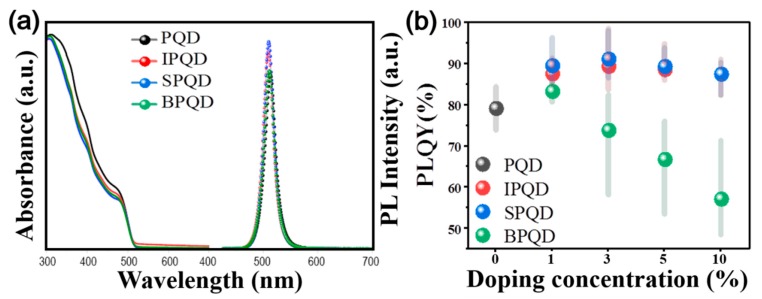
Characterization of pure and trivalent ion substituted CsPbBr_3_ perovskite QDs. (**a**) Absorbance and photoluminescence (PL) spectra, and (**b**) photoluminescence quantum yield (PLQY) value averaged from 5 measurements.

**Figure 3 nanomaterials-10-00710-f003:**
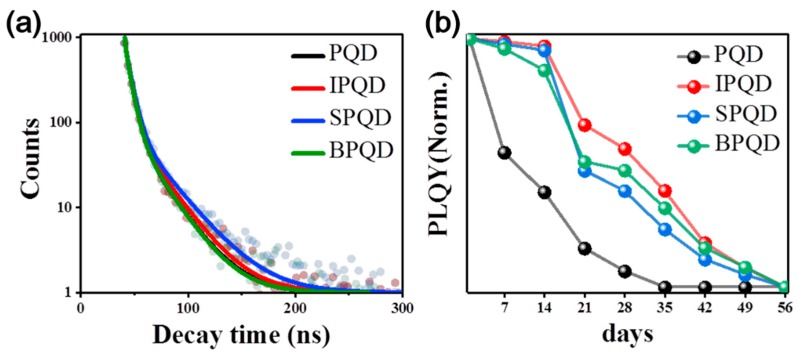
(**a**) Time-resolved PL decay with fitting curve of (**b**) stability of 0 mol%, 3 mol% In, 3 mol% Sb, and 1 mol% Bi ion substituted CsPbBr_3_ perovskite QDs.

**Figure 4 nanomaterials-10-00710-f004:**
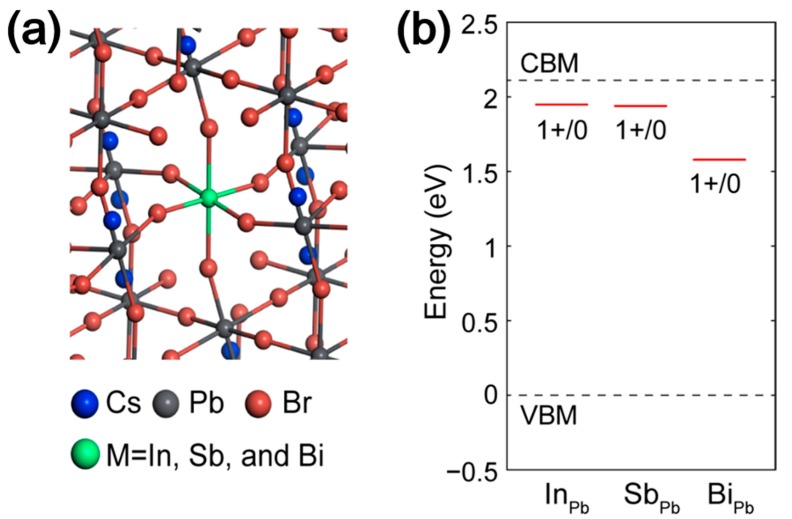
(**a**) Atomic structure simulation of CsPbBr_3_ perovskite QDs with substituted defects. (**b**) The position of the thermodynamic defect level of for In_Pb_, Sb_Pb_, and Bi_Pb_ in order.

**Table 1 nanomaterials-10-00710-t001:** Photoluminescence parameters of CsPbBr_3_ perovskite quantum dots doped with different metal ions. Each result has an average of five measurements.

Doping Concentration (%)	In			Sb			Bi		
	PLQY (%)	Peak (nm)	FWHM (nm)	PLQY (%)	Peak (nm)	FWHM (nm)	PLQY (%)	Peak (nm)	FWHM (nm)
0	81.4	513	23	81.4	513	23	81.4	513	23
1	86.8	514	23	86.8	510	23	83.0	511	23
3	88.8	509	22	91.2	510	22	72.4	508	22
5	85.8	508	23	88.0	507	23	71.2	505	22
10	82.1	504	23	82.2	501	23	57.2	499	23

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
