# Peer review of "Enhancement of Photoluminescence Quantum Yield and Stability in CsPbBr3 Perovskite Quantum Dots by Trivalent Doping"

_nanomaterials, 2020, doi:10.3390/nano10040710_

Round 1

Reviewer 1 Report

The paper titled Enhancement of Photoluminescence Quantum Yield and Stability in CsPbBr3 Perovskite Quantum Dots by Trivalent Doping from Sujeong Jung et al, it is a quite well structure paper but requires mandatory revision, aiming to broad its impact. As an overall comment, the authors must take care how they report the figures, once they are not properly put and discussed in sequential order (see page 5, figure 1! Line 182) and so on (line 201….) as well as what we expect as a supplementary information… Moreover, some sentences need to be polished and improved. See my comments below.

Title: It is aligned with the study performed.

Abstract: the abstract contains the qualitative and quantitative information required concerning the study performed.

Introduction: It is broad as required but some suggestions are given: When referring to cesium replacement of CsPbX3, it is relevant to identify the role of the fabrication process has. Please see Brites, Maria Joao et al in Ultrafast Low-Temperature Crystallization of Solar Cell Graded Formamidinium-Cesium Mixed-Cation Lead Mixed-Halide Perovskites Using a Reproducible Microwave-Based Process, ACS APPLIED ENERGY MATERIALS   Volume: 2   Issue: 3   Pages: 1844-1853   Published: MAR 2019. Moreover, it is relevant to understand the anomalous charge distribution inside quantum dots. See Panigrahi, Shrabani et al in Imaging the Anomalous Charge Distribution Inside CsPbBr3 Perovskite Quantum Dots Sensitized Solar Cells, ACS NANO   Volume: 11   Issue: 10   Pages: 10214-10221   Published: OCT 2017.

Materials and Methods: Please identify how many samples where produced and how they are reproducible? What are the average deviation concerning the characteristics of the films processed. Moreover, identify the environmental conditions where the films were characterized (humidity, temperature, atmosphere, etc.…).

Results and discussion: they are coherent with some corrections to be made. For instance. Figure 1 in page 5 (?) in b) it is missing the error bars as well in the caption a comment saying that the line is just to guide de eyes! They do not correspond to any fitting!

The equation depicted in line 216 was not deduced by the authors an so requires a reference e associated. Also, please, try to identify the deep and shallow defects and what it is their role in controlling the quantum efficiency and yield (have a look on your wrong number figure 2b). Also discuss a little the feasibility of the process and the samples reproducibility.

Conclusions: requires some improvement, according the comments made above.

References: need to be updated

Supplementary information: It is a copy of the paper? I think the authors musts just add the required information to better understand the plots they discuss in the core of the paper as well statistical data on the characteristics found as well as on the samples process.

Author Response

Dr. Myungkwan Song

Korea Institute of Materials Science

Materials Center for Energy Convergence

Surface Technology Division

797 Changwondae-ro, Seongsan-gu

Changwon-si, Gyeongsnanam-do 51508, KOREA

[Phone] +82-55-280-3686 / [FAX] +82-55-280-3570

[E-mail] smk1017@kims.re.kr

April 03, 2020

Dear Editor-in-Chief

Nanomaterials

Thank you for handling our manuscript entitled ““Enhancement of Photoluminescence Quantum Yield and Stability in CsPbBr3 Perovskite Quantum Dots by Trivalent Doping” (nanomaterials-761681)

We have carefully read the reviewers’ comments describing our work and observations as original, timely, and important. We have revised our manuscript in accordance with the reviewers’ comments. These corrections are highlighted yellow color for first reviewer and sky-blue color for second reviewer in the revised manuscript. We believe that we have satisfactorily addressed the reviewers’ technical concerns our revised manuscript.

With the incorporated changes, we hope the revised manuscript is now acceptable for publication in Nanomaterials. Should you have any further questions or comments, please let me know.

Detailed responses to the reviewers’ comments are enumerated below:

Reply to Reviewer #1

The paper titled Enhancement of Photoluminescence Quantum Yield and Stability in CsPbBr3 Perovskite Quantum Dots by Trivalent Doping from Sujeong Jung et al, it is a quite well structure paper but requires mandatory revision, aiming to broad its impact. As an overall comment, (1) the authors must take care how they report the figures, once they are not properly put and discussed in sequential order (see page 5, figure 1! Line 182) and so on (line 201….) as well as (2) what we expect as a supplementary information… Moreover, some sentences need to be polished and improved. See my comments below.

  • (1) The authors must take care how they report the figures, once they are not properly put and discussed in sequential order (see page 5, figure 1! Line 182) and so on (line 201….).
  1. First of all, we appreciate your kind comments. The sequence of figures seems to have been mixed as the ‘Materials and Methods’ session behind ‘Conclusion’ moved forward to the ‘Results and Discussion’. Finally, the sequence of figures was rearranged in sequential order according to the reviewer’s comment.
  • (2) What we expect as a supplementary information.
  1. We deeply apologize for the mis-uploading of the ‘Supplementary Information’. The supplementary information has been properly uploaded.Title: It is aligned with the study performed.Introduction: It is broad as required but some suggestions are given: (3) When referring to cesium replacement of CsPbX3, it is relevant to identify the role of the fabrication process has. Please see Brites, Maria Joao et al in Ultrafast Low-Temperature Crystallization of Solar Cell Graded Formamidinium-Cesium Mixed-Cation Lead Mixed-Halide Perovskites Using a Reproducible Microwave-Based Process, ACS APPLIED ENERGY MATERIALS   Volume: 2   Issue: 3   Pages: 1844-1853   Published: MAR 2019. Moreover, (4) it is relevant to understand the anomalous charge distribution inside quantum dots. See Panigrahi, Shrabani et al in Imaging the Anomalous Charge Distribution Inside CsPbBr3 Perovskite Quantum Dots Sensitized Solar Cells, ACS NANO   Volume: 11   Issue: 10   Pages: 10214-10221   Published: OCT 2017.
  2. Abstract: the abstract contains the qualitative and quantitative information required concerning the study performed.
  3.  
  • (3) When referring to cesium replacement of CsPbX3, it is relevant to identify the role of the fabrication process has. Please see Brites, Maria Joao et al. ACS APPLIED ENERGY MATERIALS 2, 3, (2019) 1844-1853.
  1. Thank you so much for introducing a good paper on the synthesis on FACsPbIBr perovskites using the Microwave process. Through this paper, we have been able to refer to the inherent characteristics of perovskite materials and added to the reference [21].
  • (4) It is relevant to understand the anomalous charge distribution inside quantum dots. See Panigrahi, Shrabani et al. ACS NANO 11, 10, (2017) 10214-10221.
  1. Thank you so much again for introducing a good paper on perovskite quantum dots sensitized solar cells produced using Lead-halide perovskite quantum dots. This paper is also added as a reference [9].Materials and Methods: (5) Please identify how many samples where produced and how they are reproducible? What are the average deviation concerning the characteristics of the films processed. Moreover, (6) identify the environmental conditions where the films were characterized (humidity, temperature, atmosphere, etc.…).
  2.  
  • (5) Please identify how many samples where produced and how they are reproducible?
  1. We have conducted experiments almost 10 times. The PLQY data, which is averaged by five measurements by selecting the most stable sample among them, can be found in Figure 2b. We think it has been properly reproducible by adding error bars.  Furthermore, Figure 3b is not considered appropriate to insert the error bar because it is only the highest PLQY sample for each doping substance. Therefore, the graph shown in Figure 3b is considered appropriate.
  2. And the averaged values of the PLQYs are added in Table 1 on page 4.
  3.  
  • (6) Identify the environmental conditions where the films were characterized (humidity, temperature, atmosphere, etc….).
  1. In the synthesis of perovskite quantum dots, all experiments were conducted in hood with both an average temperature of 20℃ and a humidity of 40%. The following are added to ‘Chapter 2.2 Synthesis of quantum dots’ on page 2.Results and discussion: They are coherent with some corrections to be made. For instance. (7) Figure 1 in page 5 (?) in b) it is missing the error bars as well in the caption a comment saying that the line is just to guide de eyes! They do not correspond to any fitting! (8) The equation depicted in line 216 was not deduced by the authors so requires a reference e associated. Also, please, (9) try to identify the deep and shallow defects and what it is their role in controlling the quantum efficiency and yield (have a look on your wrong number figure 2b). Also (10) discuss a little the feasibility of the process and the samples reproducibility.
  2.  
  • (7) Figure 3b in page 5, it is missing the error bars as well in the caption a comment saying that the line is just to guide de eyes!
  1. First of all, with the reviewer’s kind comment, the wrong figure numbering was corrected from Figure 1b to Figure 3b on page 5.
  2. As mentioned in question 5, Figure 3b, which shows the stability of PLQY over days, is the data measured using samples of the highest PLQY for each doping substance, and it is not appropriate to insert the error bar.
  • (8) The equation depicted in line 216 was not deduced by the authors so requires a reference associated.
  1. The equation on page 6 was induced by [Ref. 39, (Phys. Rev. Lett. 102 (2009) 016402)] with the equation at 2.4 calculation on page 3. In addition, with the reviewer’s kind comment, the equation on page 6 mentioned by the reviewer is cited as [Ref. 39].
  • (9) Also, please, try to identify the deep and shallow defects and what it is their role in controlling the quantum efficiency and yield (have a look on your wrong number figure 2b).
  1. First of all, the wrong figure numbering was corrected from Figure 2b to Figure 4b on page 6.
  2. In our study, the three defects, namely BiPb, SbPb, and InPb, are donors and BiPb is deeper than SbPb and InPb because its thermodynamic defect level (TDL) is lower in energy than the others. In accordance with the Shockley-Read-Hall (SRH) model, the overall rate of SRH recombination increases as the TDL approaches the mid gap. Therefore, BiPb is the most detrimental for the quantum yield comparing to SbPb and InPb.
  • (10) Discuss a little the feasibility of the process and the samples reproducibility.
  1. As mentioned in question 5, the CsPbBr3 perovskite quantum dots, which is synthesized according to the trivalent doping substance, is the average value of the PLQY obtained by experimenting within 10 times and is deemed sufficiently reliable. Conclusions: requires some improvement, according the comments made above.
  2. References: need to be updated
  3.  
  • A) Two papers recommended by the kind advice of the reviewer were added at the reference [9], [21], respectively.
  • Supplementary information: It is a copy of the paper? I think the authors musts just add the required information to better understand the plots they discuss in the core of the paper as well statistical data on the characteristics found as well as on the samples process.
  • A) We apologize once again deeply for the mis-uploading of the ‘Supplementary Information’. The supplementary information has been properly uploaded.
  •  

Reviewer 2 Report

The manuscript of S. Jung et al. is devoted to study of a role of CsPbBr3 nanocrystal (NC) doping by In, Sb, and Bi trivalent ions on PLQY and stability of NCs. The topic is important because of promising optical and electronic properties of metal halide perovskites for photovoltaic and light emitting devices. The authors found a remarkable increase in PLQY at certain doping ion concentrations and explained the improvements on the basis of literature data and own calculations. The work is written well and can be published practically without any modifications, except some minor mistakes which require to be corrected:

line 75: should be "All reagents..."
line 215: should be "of defects" but not "the of defects".

line 249: it is better to write "fermi level rise" instead of "an increase in Fermi level".

Author Response

Dr. Myungkwan Song

Korea Institute of Materials Science

Materials Center for Energy Convergence

Surface Technology Division

797 Changwondae-ro, Seongsan-gu

Changwon-si, Gyeongsnanam-do 51508, KOREA

[Phone] +82-55-280-3686 / [FAX] +82-55-280-3570

[E-mail] smk1017@kims.re.kr

April 03, 2020

Dear Editor-in-Chief

Nanomaterials

Thank you for handling our manuscript entitled ““Enhancement of Photoluminescence Quantum Yield and Stability in CsPbBr3 Perovskite Quantum Dots by Trivalent Doping” (nanomaterials-761681)

We have carefully read the reviewers’ comments describing our work and observations as original, timely, and important. We have revised our manuscript in accordance with the reviewers’ comments. These corrections are highlighted yellow color for first reviewer and sky-blue color for second reviewer in the revised manuscript. We believe that we have satisfactorily addressed the reviewers’ technical concerns our revised manuscript.

With the incorporated changes, we hope the revised manuscript is now acceptable for publication in Nanomaterials. Should you have any further questions or comments, please let me know.

Detailed responses to the reviewers’ comments are enumerated below:

Reply to Reviewer #2

The manuscript of S. Jung et al. is devoted to study of a role of CsPbBr3 nanocrystal (NC) doping by In, Sb, and Bi trivalent ions on PLQY and stability of NCs. The topic is important because of promising optical and electronic properties of metal halide perovskites for photovoltaic and light emitting devices. The authors found a remarkable increase in PLQY at certain doping ion concentrations and explained the improvements on the basis of literature data and own calculations. The work is written well and can be published practically without any modifications, except some minor mistakes which require to be corrected:

line 75: should be "All reagents..."
line 215: should be "of defects" but not "the of defects".

line 249: it is better to write "fermi level rise" instead of "an increase in Fermi level".

  • A) First of all, we sincerely thank you for your deep interest in our research and for your favorable assessment of our results.    On the basis of the above explanations, we now hope our revised manuscript would prove satisfactory for all readers and referees, and will be accepted for publication in Nanomaterials. Should you have any further questions or comments, please let me know.
  •  
  •  
  •  
  • In accordance with the reviewer’s kind comments, we have properly modified the above three parts recommended for correction.

Sincerely,

Dr. Myungkwan Song

Round 2

Reviewer 1 Report

T&he paper was properly revised but still remain some need of english polish once some sentence are not properl!